# Gram-Positive Pneumonia: Possibilities Offered by Phage Therapy

**DOI:** 10.3390/antibiotics10081000

**Published:** 2021-08-18

**Authors:** Lucía Fernández, María Dolores Cima-Cabal, Ana Catarina Duarte, Ana Rodríguez, María del Mar García-Suárez, Pilar García

**Affiliations:** 1Instituto de Productos Lácteos de Asturias (IPLA-CSIC), Paseo Río Linares s/n, 33300 Villaviciosa, Asturias, Spain; lucia.fernandez@ipla.csic.es (L.F.); catarina.leal@ipla.csic.es (A.C.D.); anarguez@ipla.csic.es (A.R.); 2DairySafe Group, Instituto de Investigación Sanitaria del Principado de Asturias (ISPA), 33011 Oviedo, Asturias, Spain; 3Escuela Superior de Ingeniería y Tecnología (ESIT), Universidad Internacional de la Rioja (UNIR), Av. de la Paz, 137, 26006 Logroño, La Rioja, Spain; dolores.cima@unir.net

**Keywords:** pneumonia, antibiotic resistance, *Streptococcus pneumoniae*, *Staphylococcus* *aureus*, new therapies, phage therapy, endolysins

## Abstract

Pneumonia is an acute pulmonary infection whose high hospitalization and mortality rates can, on occasion, bring healthcare systems to the brink of collapse. Both viral and bacterial pneumonia are uncovering many gaps in our understanding of host–pathogen interactions, and are testing the effectiveness of the currently available antimicrobial strategies. In the case of bacterial pneumonia, the main challenge is antibiotic resistance, which is only expected to increase during the current pandemic due to the widespread use of antibiotics to prevent secondary infections in COVID-19 patients. As a result, alternative therapeutics will be necessary to keep this disease under control. This review evaluates the advantages of phage therapy to treat lung bacterial infections, in particular those caused by the Gram-positive bacteria *Streptococcus pneumoniae* and *Staphylococcus aureus*, while also highlighting the regulatory impediments that hamper its clinical use and the difficulties associated with phage research.

## 1. Introduction

Pneumonia is a disease that arises when a pathogen reaches the lower respiratory tract, overcomes the host defense system and damages the pulmonary parenchyma. Although non-infectious forms of this pathology do exist, the so-called idiopathic interstitial pneumonia, they are out of the scope of this review. Infectious pneumonia can be mild, but it may also progress to a severe, life-threatening condition, depending on the host characteristics and the virulence of the pathogen. In adults, the risk of acquiring bacterial pneumonia increases with age, stays in long-term care facilities, and comorbidities, such as stroke or neurological deterioration [1]. In young children, especially in low-income countries, childhood wasting and household air pollution are the underlying risk factors for morbidity and mortality due to this illness. It must be noted that after the introduction of Hib and pneumococcal conjugate vaccines (against *Haemophilus influenzae* type b and *Streptococcus pneumoniae*, respectively), pneumonia mortality rates have significantly decreased [2]. However, this disease remains, to this day, a major cause of death in children, the elderly and the immunocompromised.

Pneumonia can be caused by a wide variety of germs, including bacteria, viruses and fungi. Moreover, the recent advances in molecular detection techniques have revealed that, in some cases, different microorganisms (e.g., a virus and a bacterium) can co-exist to produce the disease. Depending on the environment where the pathogen is acquired, this illness is often broadly classified into community-acquired pneumonia (CAP) and hospital-acquired pneumonia (HAP). The latter is also referred to as nosocomial pneumonia, and includes ventilator-associated pneumonia (VAP), which is defined as pneumonia occurring >48 h after endotracheal intubation.

So far, at least 26 viruses have been associated with CAP in both children and adults. In adults, influenza viruses, rhinoviruses and coronaviruses are responsible for a third of pneumonia cases, while respiratory syncytial virus, rhinovirus, human metapneumovirus, human bocavirus and parainfluenza viruses are the main agents identified in children. HAP may also be caused by viral pathogens, resulting in a similar death rate to that observed for bacterial infections [3].

Regarding fungi, there are several yeasts and molds (*Pneumocystis jiroveci*, *Cryptococcus neoformans*, *Aspergillus* and *Fusarium*) that can also colonize the respiratory tract and cause disease [4] (Figure 1). Over the last decades, the incidence of fungal pneumonia has risen in highly immunosuppressed patients, such as those affected by HIV/AIDS, cancer, solid organ transplants and other chronic pulmonary diseases, such as cystic fibrosis.

The major etiological agents of bacterial CAP are *S. pneumoniae* and *H. influenzae*, although atypical bacteria, such as *Mycoplasma pneumoniae*, *Chlamydia pneumoniae* and *Legionella pneumoniae*, are also responsible for an important number of cases [5]. In that sense, it must be noted that the term atypical pneumonia is somewhat inaccurate because these microbes are not an uncommon cause of CAP in adults. On the other hand, nosocomial pneumonia is mainly caused by Gram-negative bacteria, including *Pseudomonas aeruginosa*, extended-spectrum β-lactamase-positive (ESBL+) Enterobacteriaceae, multidrug resistant (MDR) *Acinetobacter baumannii*, *Stenotrophomonas maltophilia* as well as the Gram-positive methicillin-resistant *Staphylococcus aureus* (MRSA) [6].

Since February 2020, antibiotic use has risen dramatically worldwide due to SARS-CoV-2 infections. Indeed, it is estimated that 70–97% of hospitalized patients with COVID-19 receive antibiotic therapy [7]. For instance, patients presenting symptoms associated with a respiratory infection are often treated with antibiotics before infection with SARS-CoV-2 is confirmed. Additionally, antibiotics are sometimes prescribed as a preventive measure against secondary bacterial infections in severe COVID-19 patients, in which hospitalization and/or intubation increased the risk of such infections. The main bacteria associated with SARS-CoV-2 secondary infections and co-infections are *S. pneumoniae* and *S. aureus*, which also happen to be the most prevalent Gram-positive microbes causing pneumonia [8]. These two species are major colonizers of the upper respiratory tract, which facilitates their access to lower parts of the respiratory system, and they are in the WHO list of pathogens for which new antibiotics are urgently needed (https://www.who.int/news/item/27-02-2017-who-publishes-list-of-bacteria-for-which-new-antibiotics-are-urgently-needed (accessed on 15 June 2021)). In this context, the recent approval of phage therapy by the FDA for the treatment of COVID-19 patients with secondary bacterial infections has once again put this strategy in the spotlight as a feasible alternative to combat antibiotic-resistant pathogens.

This review summarizes the current lines of research aimed at improving the diagnosis, prevention and treatment of Gram-positive pneumonia, with special emphasis on the therapeutic potential of phages and their derived proteins.

## 2. Disease Incidence and Mortality

The World Health Organization (WHO) has estimated that 450 million cases of pneumonia are recorded globally each year, with a death toll of approximately four million people. Moreover, pneumonia accounts for 15% of all deaths in children under 5 years of age, killing 808,694 children in 2017 [9]. However, the incidence and etiology of pneumonia are strongly correlated with its geographic distribution. Poverty, socioeconomic factors and malnutrition influence the incidence and outcome of CAP in some countries, such as Latin America and the Caribbean region, where *S. pneumoniae* has an incidence of 24–78% [10]. It must be noted that HAP, including VAP, represented almost 22% of all nosocomial infections from 2002 to 2003, which highlights the importance of pneumonia in the hospital environment. Moreover, VAP exhibited high morbidity and mortality rates, with the latter ranging between 30 and 70% [11]. 

With an incidence of 27.3%, *S. pneumoniae* is the major causative agent of CAP. This bacterium is a colonizer of the upper respiratory tract, especially in children, in whom its prevalence is around 20–40%. This is significant since carriage of this bacterium is known to increase the risk of *S. pneumoniae* infections, including pneumonia. Indeed, before the introduction of a 7-valent pneumococcal conjugate vaccine (PCV7) for children, 63,000 cases of invasive pneumococcal disease occurred each year in the US. These cases were caused by both vaccine serotypes (serotypes covered by PCV7) and non-vaccine serotypes (serotypes not covered by PCV7) of pneumococci [12]. 

The Gram-positive pathogen *S. aureus* is the etiological agent of 1.7% cases of CAP, of which 0.7% involve methicillin-resistant (MRSA) strains. Interestingly, introduction of the pneumococcal conjugate vaccine in Germany was followed by a change in the etiological agents causing invasive disease in children. Indeed, the most commonly detected pathogens in pediatric parapneumonic pleural effusions/emphysema patients were non-vaccine serotypes of *S. pneumoniae*, *S. pyogenes* and *S. aureus* [13]. *S. aureus* is responsible for 28% of all nosocomial pneumonia infections [14], still ranking as the number one cause in the US, many EU countries, South Korea and Singapore. The risk of developing an infection by *S. aureus* is high partly because this bacterium is a member of the microbiota of the anterior nares. Indeed, it is considered that approximately 30% of individuals are colonized with this bacterium, and, worryingly, the rate of nasal colonization with MRSA strains is increasing. Moreover, in addition to antibiotic resistance, many staphylococcal strains produce virulence determinants, which facilitate colonization and evasion of the immune system, resulting in high mortality rates (56–75%) [15]. 

## 3. Diagnosis of Bacterial Pneumonia

Traditionally, the diagnosis of pneumonia has been carried out through culture-based methods starting from different types of clinical samples, such as blood, sputum, pleural fluids, etc. However, culture-dependent techniques have several drawbacks. For example, test results take between two to three days to come back. Moreover, a positive diagnosis can only be achieved if the infectious agent remains alive in the host tissues, which sometimes does not happen if the patient has previously been administered antibiotics. Microscopy and serology are also commonly used for the diagnosis of pneumonia, especially for the detection of atypical bacteria (*Mycoplasma* spp. and *Chlamydia* spp.). Additionally, methods based on specific antigen–antibody reactions have represented a significant advance in bacterial diagnosis of this disease. For example, the application of direct immunofluorescence techniques to respiratory samples from pneumonia patients has enabled the development of rapid tests for *S. pneumoniae* [16]. 

The development of techniques for the rapid detection of certain pneumococcal antigens in urine has led to a significant improvement compared to traditional cultivation methods [17]. For instance, pneumolysin, an important *S. pneumoniae* virulence factor, has been confirmed as a relevant marker for diagnosis of pneumococcal pneumonia. Unfortunately, the immunoassays required to detect this antigen are not commercially available at the moment. Currently, there is a commercial diagnostic test for the detection of pneumococcal polysaccharide C in urine (*S. pneumoniae* Binax Now^®^). This test is used in adults, in which diagnosis of pneumococcal pneumonia has acceptable specificity and sensitivity (95–80%). In children, however, this technique has low specificity and cannot discern between sick and asymptomatic carrier children [18]. Consequently, pediatricians encounter many difficulties when trying to identify the etiological agent of CAP in their patients. In such cases, due to the absence of specific and fast diagnostic tools, pneumonia diagnosis is primarily based on chest radiography, and antibiotic treatment is frequently empirical. 

In recent times, traditional diagnostic methods have been complemented by molecular methodologies, specifically nucleic acid detection tests (NATs), such as polymerase chain reaction (PCR). These techniques, especially PCR, have represented a notable advance in terms of speed and specificity in the diagnosis of pneumonia on various clinical samples (blood, pleural fluid and bronchoalveolar lavage). For example, the utilization of NATs has allowed the identification of different CAP-causative pathogens. These tests have been particularly advantageous in the case of *M. pneumoniae*, *Legionella* spp. and *C. pneumoniae*. In contrast, the advantages of NATs over more conventional techniques for the detection of pneumococcus are not so clear due to the difficulty of distinguishing between infected patients and carriers when analyzing respiratory samples [19]. However, researchers continue to develop new tests that try to solve this problem. For instance, Bjarnason et al. [20] recently developed a real-time PCR test using samples from the oropharynx of adults suffering from pneumonia. By applying different cut-off values, the authors could successfully distinguish between patients and carriers. Indeed, this test achieved a sensitivity of 87% and a specificity of 79% for the detection of *S. pneumoniae*. Therefore, this method can be a useful tool for confirming a diagnosis established by other methods, especially in patients who cannot provide samples from the lower respiratory tract [20].

It is worth noting that 25% of CAP patients have polymicrobial (or mixed) infections, in which several microorganisms cause the disease. Indeed, the study carried out by the Pneumonia Etiology Research for Child Health (PERCH) project in Africa and Asia, published in 2019, corroborates this tendency to observe mixed infections [21]. With this in mind, several multiplex PCR tests have been developed in order to improve the etiological diagnosis of the illness [22]. In this context, perhaps the most relevant tests are those used for the analysis of throat and nasal swabs that include a panel of viruses and bacterial pathogens to increase etiological yield in hospitalized children suffering from CAP [23]. Lee et al. [24] also have developed a multiplex PCR protocol that allows identification of pneumonia-causing pathogens, as well as their main resistance determinants, in adults hospitalized in intensive care units [24].

Despite its many advantages, PCR has the disadvantage of requiring specialized equipment and qualified personnel. In contrast, another technique based on nucleic acid amplification, loop-mediated (LAMP) isothermal amplification [25], is less expensive than conventional PCR, can be carried out by non-specialized personnel and, above all, does not require specific equipment. This method can be applied for the detection of both viruses and bacteria, since both DNA and RNA can be amplified. So far, this technique has proven to be useful for the detection of *M. pneumoniae* [26] and *S. pneumoniae* [27], among others.

More recently, more sophisticated methods, such as metagenomic next-generation sequencing (mNGS), have been presented as a potential solution for diagnosis of polymicrobial pneumonia. This strategy might be especially helpful in cases of severe pneumonia in children, where obtaining an accurate diagnosis is particularly challenging [28]. This technique provides increased sensitivity when detecting the pathogenic microorganisms that cause the disease, giving additional information about the strain and helping to identify new pathogens. However, it should be noted that this technique is complex and expensive and is not widely applied at this time.

Another novel, promising technique involves the use of microfluidic chips, which are small-size platforms made of different materials that can integrate different diagnostic tests, such as PCR or LAMP, thereby reducing the reaction time, and, consequently, favoring decentralized analysis [29]. There are already some models specifically designed for the diagnosis of pneumonia. For example, a microchip platform for point-of-care testing of *S. pneumoniae* and *M. pneumoniae* has been developed recently. The clinical sensitivity and specificity of this platform was evaluated using 63 randomly selected oropharyngeal swabs and bronchoalveolar lavage fluid specimens from children. This platform was shown to be more sensitive, faster, and as specific as conventional PCR tests against *S. pneumoniae* [30]. 

## 4. Preventive and Therapeutic Strategies against Gram-Positive Pneumonia

### 4.1. Pneumonia Prevention: Impact of Vaccination

As mentioned above, bacterial pneumonia, including that caused by Gram positives, mainly affects children and the elderly. In these population groups, several factors have an impact on the morbidity and mortality due to this illness. For example, an adequate nutrition and exclusive breastfeeding contribute to reducing the incidence of pneumonia in children and infants. In general, avoiding air pollution is also considered an important factor, given that an increase in pneumonia hospitalization rates has been observed in contaminated environments [31]. Nonetheless, immunization remains the most effective way to prevent this disease. 

The polyvalent pneumococcal polysaccharide vaccine was licensed for its use in the United States in the 1980s. This vaccine contained 23 capsular polysaccharides and its use was encouraged for adults > 65 years of age [32]. Later on, in 2000, a 7-valent pneumococcal conjugate vaccine (PCV7), designed against those serotypes that most frequently cause invasive pneumococcal disease, was introduced for use in children. Currently, this vaccine is being administered in at least 130 countries. Since 2009, two pneumococcal conjugate vaccines are also available, PCV13 (Prevenar^TM^ 13) and PCV10 (Synflorix^TM^). Widespread use of these vaccines has successfully reduced invasive pneumococcal disease, although the prevalence of some serotypes not covered by the vaccines has been gradually increasing due to serotype replacement. 

Several vaccines against *S. aureus* are currently under development, some of which have successfully been assayed in the preclinical phase. Some of these vaccines have already entered the clinical phases of development, but unfortunately none have reached phase III clinical trials yet [33].

### 4.2. Present and Future Therapeutic Strategies

Regarding the treatment of pneumonia caused by Gram-positive bacteria, several options of antibiotic therapy are available at the moment (Table 1). However, the need to tackle the spread of antibiotic resistance has also led to the design of new strategies, several of which are currently under development (Table 1).

As mentioned previously, pneumonia is mostly diagnosed on the basis of its clinical signs and symptoms. For this reason, it is most frequently treated with antibiotics covering the most likely causative pathogens. International health authorities recommend starting with a fluoroquinolone (levofloxacin or moxifloxacin) or a β-lactam (amoxicillin, ampicillin, amoxicillin-clavulanate, ampicillin-sulbactam, cefotaxime or ceftriaxone) plus a macrolide (azithromycin or clarithromycin). Patients with moderate *S. pneumoniae* pneumonia may respond to oral amoxicillin, whereas severe pneumonia may need intravenous ceftriaxone, cefotaxime or amoxicillin-clavulanic acid [34]. When the presence of *S. aureus* is confirmed, treatment generally includes penicillins (oxacillin and/or flucloxacillin). However, MRSA strains require additional treatment with linezolid or vancomycin, although nephrotoxicity and low penetration in lung tissues are the main drawbacks for vancomycin. As both vancomycin and linezolid have similar efficacy, the election of one or the other is based on patient tolerance, renal function and intravenous access (linezolid is available in oral form whereas vancomycin is preferred for patients using selective serotonin reuptake inhibitors). Recently, a broad-spectrum activity cephalosporin, ceftaroline fosamil, has been approved in the United States and Europe for the treatment of adults with moderate-to-severe CAP. Furthermore, the US Food and Drug Administration (FDA) (19 August 2019) and the European Medicines Agency (26 May 2020) approved lefamulin for the treatment of CAP as it has demonstrated efficacy against *S. aureus*, beta-hemolytic and viridans group streptococci, coagulase-negative staphylococci, *Enterococcus faecium*, *S. pneumoniae*, *H. influenzae*, *M. pneumoniae*, *C. pneumoniae*, *L. pneumophilia* and *Moraxella catarrhalis*.

One of the main problems of antibiotics is that they readily select resistant variants. As a result, it is necessary to constantly develop new ones that are still effective against non-susceptible strains or, alternatively, design new antibacterial strategies. Like antibiotics, some of these therapies also intend to kill the pathogen or inhibit its growth within the host. For instance, antimicrobial peptides represent an interesting alternative to conventional antibiotics, although bacterial resistance has already been observed in vitro [35] and it remains to be determined if resistance selection also occurs in vivo. Co-administration of AMPs with an exogenous surfactant allowed a proper distribution of the peptides in the lung. Four AMPs (CATH-1, CATH-2, CRAMP and LL-37) suspended in bovine lipid-extract surfactant (BLES) were successfully evaluated in vitro in a surfactant–AMP mixture against MRSA and *P. aeruginosa* [36]. Bacteriophages are also being rediscovered as promising antibacterial agents, and as a source of novel antimicrobial enzymes or enzybiotics. Phage-derived antibacterials will be the focus of the next section of this review, and will be discussed more in depth.

In other cases, the objective is not to destroy the pathogen, but rather to limit its ability to damage the host. It is well known that disease severity is caused by the cell damaging and inflammatory effects of some toxins secreted by infectious pathogenic bacteria. For this reason, some strategies are aimed at neutralizing these toxins and their effects. In this context, several molecular targets for antibody neutralization have been proposed for *S. pneumoniae* infections, including pneumolysin PLY and the choline-binding protein PspA [37], although clinical trials have not been performed yet. In turn, neutralizing antibodies against staphylococcal toxins have been successfully tested in a variety of trials. An example is the human monoclonal IgG1 antibody AR-301 (Salvecin^TM^, Aridis Pharmaceuticals, Inc, San Jose, CA, USA), which has been evaluated in clinical trials as a complement to antibiotic therapy for *S. aureus* pneumonia in ICU patients [38]. The mAb MEDI4893 (MedImmune, LLC, Gaithersburg, MD, USA) is also undergoing clinical trials. Further examples include 2A3 and its variant LC10, which reduce disease severity in a murine model of *S. aureus* pneumonia [39], and mAb LTM14, which provides protection against *S. aureus* [40]. Regarding antibodies against leukocidins, the mAb ASN-1 was found to bind four cytotoxins—LukSF-PV (Panton-Valentine leucocidin), LukED, HlgAB and HlgCB—in addition to Hla, whereas the mAb ASN-2 neutralized the leukotoxin LukGH. Therefore, both were further combined to achieve a broader range of neutralization (ASN-100, Arsanis, Inc., Waltham, MA, USA) [41]. The next generation of mAbs have been designed to include novel targets, such as the *agr* components from *S. aureus* [42]. Liposomes are also being explored as decoys to capture pore-forming toxins. These formulations improved the therapeutic efficacy of antibiotics in bacteremia models [43]. A different strategy to avert the virulence of the pathogen is to prevent toxin production or assembly by using natural compounds, such as flavonoids (baicalin, morin hydrate) [44]. Finally, another approach for controlling toxin-related damage is the inhibition of host cell receptors to avoid binding and uptake of these toxins. This is the case of the hydroxamate inhibitor GI254023X, which inhibits the Hla receptor protein ADAM10, thereby inhibiting binding of the toxin to the eukaryotic cell and, consequently, minimizing toxicity and lesion size in animal models [45].

Another interesting approach to combat bacterial pneumonia is immunomodulation, i.e., use of the ability of the immune system to fight against microbes, while reducing the inflammatory response to infection. Some immune modulators include defense-regulating peptides and agonists of immune components, such as Toll-like receptors and NOD-like receptors, and even certain microbial signaling molecules (N-acyl homoserine lactones and cyclic nucleotides) [46].

Last but not least, it is worth noting that the growing understanding of the relationship between having a balanced microbiota and human health is also opening the door to new strategies for disease treatment. However, unlike the gut microbiota, very little is known about the lung microbiota in healthy individuals. Therefore, it would be interesting to identify which commensal microorganisms are beneficial for protection against lung pathogens.

## 5. Phage Therapy

### 5.1. Bacteriophages as New Weapons against Pneumonia-Causing Bacteria

Phage therapy is a promising option for the treatment of Gram-positive pneumonia, especially when caused by antibiotic-resistant pathogens. Bacterial viruses offer some interesting advantages over other types of antimicrobials, such as their abundance in nature, their ability to multiply during treatment, their safety for human health and the environment and the lack of cross-resistance with most antibiotics [47]. 

In the case of *S. pneumoniae*, several bacteriophages infecting this species have been identified [48], but due to the abundance of temperate phages and their presence in most clinical isolates, phage therapy research has focused mostly on the use of endolysins (see below). 

In contrast, there are some successful results concerning the use of phages to treat *S. aureus* pneumonia in animal models of infection. For instance, Oduor et al. [49] found that phages were even more effective than clindamycin to treat hematogenous multidrug resistant pneumonia in mice. Similarly, phage therapy is also a promising alternative and/or complementary strategy for the treatment of *S. aureus* VAP. Indeed, administration of intravenous teicoplanin or a cocktail of four phages to infected mice increased survival to 50% and 58%, respectively [50]. Moreover, nebulized bacteriophages reduced lung bacterial burdens and improved survival in infected rats both as a prophylaxis and as a treatment for VAP [51,52]. A mixture containing three lytic myoviruses infecting *S. aureus*, AB-SA01, also has been recently evaluated in a mouse model of acute pneumonia. The effectivity of AB-SA01 was similar to that of vancomycin [53]. On the basis of these results, phase I/II clinical trials concerning the treatment of several staphylococcal infections have been initiated. However, there is no specific trial regarding pneumonia treatment for the time being. Finally, a case study reported the treatment of a cystic fibrosis patient with the Pyophage preparation (five phages against *S. aureus*, *S. pyogenes*, *P. mirabilis*, *P. vulgaris*, *P. aeruginosa* and *E. coli*) and the *S. aureus* phage Sb-1. Importantly, no adverse events were observed after phage application with a nebulizer. Furthermore, treatment resulted in a drastic reduction in bacterial cell counts, although total eradication was not achieved [54].

Despite the potential benefits of phage therapy, there are also some difficulties that must be overcome in order to promote its clinical use. One such challenge relates to choosing the most appropriate method of phage delivery (Figure 2). For example, aerosolized phages are more effective than those used by intravenous administration, as the latter would be more likely to stimulate the production of neutralizing antibodies [55]. Prazak et al. [52] found that administration of a combination of aerosolized phages and intravenous phages rescued 90% of rats with VAP. On the other hand, the immunostimulation caused by phages might be counterproductive in the already stressed lungs of pneumonia patients. Dufor et al. [56] found that intranasal application of two *E. coli* phages to healthy mice promoted a weak increase in antiviral cytokines (gamma interferon and interleukin-12) and chemokines in the lungs, whereas no over-inflammation was observed following phage treatment of the infected animals. It remains to be determined if this would also be the case for phages infecting *S. aureus* and *S. pneumoniae*. 

Unfortunately, data regarding the use of phage-derived products for pneumonia treatment in humans remain scarce. One of the few studies in this field was described by the authors in [57], who evaluated the efficacy of phages in pneumonia patients caused by different pathogens. The treatment was successful in more than 80% of cases, demonstrating the therapeutic potential of the selected phages. 

### 5.2. Phage-Derived Lytic Proteins

In addition to phages themselves, phage lysins are also valuable weapons to fight against lung infections (reviewed by [58]). These phage-encoded proteins hydrolyze the peptidoglycan from the bacterial cell wall and display interesting properties, such as the lack of bacterial resistance development together with their high antimicrobial effect, when added exogenously to Gram-positive bacteria. To date, several endolysins have been characterized and successfully tested against *S. pneumoniae*, both in vitro and in vivo. Notably, lysin Cpl-1 has shown therapeutic potential in different animal models of infection [59], sometimes in combination with antibiotics [60] or with another lysin (Pal) [61]. Similarly, lysin Cpl-7 exhibits killing activity against several bacteria [62], which may be improved even further by inverting the net charge of its cell-wall-binding domain [63]. Indeed, the chimeric proteins Cpl-711 and PL3 turned out to be more effective than their parental lytic proteins in animal models of nasopharyngeal colonization [64]. Moreover, these proteins showed a synergistic lytic activity in different assays, both in vitro (including biofilm degradation) and in vivo in an adult zebrafish model of pneumococcal infection [65]. Synergism was also observed for a combination of Cpl-711 and two antibiotics (amoxicillin and cefotaxime) against multidrug-resistant *S. pneumoniae* strains. The increase in effectivity resulting from combining the endolysin and cefotaxime was even confirmed in vivo using sepsis infection models in mice and zebrafish [66]. With the aim of improving its activity, chimeric lysin ClyJ was modified by shortening its linker, which led to increased lytic activity in vivo (20-fold) and reduced cytotoxicity [67]. Remarkably, a variant of this protein, ClyJ-3m, which remains a monomer after binding choline, exhibited even higher bactericidal activity and improved the pharmacokinetic properties, such as a lesser clearance by the immune system [68]. Additionally, two new streptococcal endolysins (23TH_48 and SA01_53) have been recently identified in the oral microbiome. One of them, endolysin 23TH_48, encoded by a phage infecting *S. infantis*, exhibits lytic activity against several *S. pneumoniae* isolates [69].

There are also many phage lysins and chimeric proteins effective against *S. aureus* [70], although most of them have yet to be tested for the treatment of respiratory infections. However, they have been used for nasal decolonization, which is very useful for preventing *S. aureus* infections, especially those that occur after surgery [71]. Regarding the treatment of staphylococcal pneumonia, a study carried out by [72] in a mouse infection model has shown promising results. These authors found that a single intranasal administration of endolysin SAL200 was enough to obtain a survival rate of 90–95% in animals previously infected with *S. aureus*. The efficacy of endolysins in combination with other therapeutic substances for the treatment of pneumonia in mice has also been tested. For example, the combination of endolysin LysGH15 and apigenin, which displays anti-inflammatory activity, was evaluated in a mouse *S. aureus* pneumonia model, proving to be more effective than LysGH15 or apigenin individually [73]. Although *S. aureus* endolysins have not been tested for treating pneumonia in humans, there are several ongoing clinical trials for their use against other staphylococcal infections [70].

Regarding the delivery of endolysins, there are still few studies. Recently, endolysin Cpl-1 was loaded into chitosan nanoparticles [74] and successfully tested in an animal infection model [75]. Kaur et al. [76] also reported a delivery system based on chitosan-alginate for endolysin LysMR-5. An important advance in this field has been the delivery of endolysins by nebulization. For instance, Wang et al. [77] showed that Cpl-1 remained stable after mesh nebulization, but lost activity during jet nebulization. This result opens new possibilities for the development of therapeutic products that can be successfully administered by inhalation.

Before phage lysins can be widely used as therapeutics, it will be necessary to demonstrate that their administration to humans is safe. In this context, several studies have examined the preclinical safety and toxicity of these proteins, and have not found significant signs of potential toxicity. For instance, no change in gene expression was observed in human macrophages and pharyngeal cells exposed to the endolysins Pal and Cpl-1. Likewise, the pro-inflammatory cytokine levels remained constant, and complement activation was not detected in animals injected with these proteins. The IgG levels increased for the first 30 days while the IgE levels remained stable [78].

## 6. Conclusions and Perspectives

Despite the significant contribution of vaccines and antibiotics towards the decrease in morbidity and mortality due to bacterial pneumonia, this disease remains a serious threat to vulnerable population groups. One of the main factors involved in this problem is the relentless spread of antibiotic-resistant strains of different pathogenic bacteria. *S. pneumoniae* and *S. aureus*, the main agents causing CAP and HAP, respectively, are no exception to this trend. Therefore, it is necessary to implement new strategies to combat this illness. There already are several promising alternatives under development, including the use of natural compounds, antibodies, antimicrobial peptides, immunomodulators and bacteriophages. Perhaps, the heightened need to manage secondary bacterial infections due to the coronavirus pandemic may accelerate the introduction of these novel strategies in order to substitute and complement antibiotics. 

Over the last decade, the research community has been trying to develop phage therapy in order to bring new possibilities for the treatment of infectious diseases caused by antibiotic-resistant pathogens. Unfortunately, many aspects of phage therapy and its potential applications are still underexplored. Even so, the administration of phages as therapeutics is currently being considered given the critical situation caused by the pandemic. Remarkably, the FDA has approved the use of phages in COVID-19 patients with secondary infections, including critical patients with pneumonia (https://clinicaltrials.gov/ct2/show/NCT04636554 (accessed on 15 June 2021)). Some studies even point out that phages can exhibit antiviral properties by helping to boost immunity against viral pathogens [79]. Nonetheless, the use of phages to combat bacterial infections still has many unknown aspects, such as how these antimicrobials work within the human body. For instance, data regarding the relationship between phage therapy efficacy and the immunity of the patient are scarce. In this context, Roach et al. [80] found that neutrophil–phage synergy is essential for treatment success in mice. It is actually known that bacteriophages can induce cytokine production in immune cells, thereby helping to reduce infection despite the fact that specific antibodies may target the phage particles [81]. 

Other issues are related to the development of methodologies that facilitate phage application in hospitals. In this regard, a strain susceptibility test will be necessary in order to design the most appropriate treatment for each patient. This will require growth of the pathogen and subsequent analysis of phage susceptibility, a process that may take more time than that used for antibiotic sensitivity testing. In this context, the project Phago Flow intends to solve this problem by developing a phagogram, a technique akin to the antibiogram but adapted to phages (https://www.phagoflow.de/en/phagogram/ (accessed on 15 June 2021)). Similarly, the Institut Pasteur has created the Viral Host Range database (VHRdb) (https://viralhostrangedb.pasteur.cloud// (accessed on 15 June 2021)), which gathers data generated by scientists from all over the world documenting the host range of their phages. Recently, Pherecydes Pharma has been awarded funding to carry out the EU project PhagoPROD, which will enable the company to implement all the necessary requirements for phage production in accordance with Good Manufacturing Practices (GMP), while ensuring the quality, safety and efficacy of these products [82]. In turn, other ongoing studies intend to increase the bioavailability of phage-derived products, for example, by encapsulating endolysins in chitosan nanoparticles to treat pneumonia [74], and to improve the yield of production systems, such as phage protein expression in plants [83]. 

Currently, most of the phage products used in the hospitals of Western countries are produced as magistral preparations, which are formulated on demand as personalized medicines. This is a drawback for companies that need to have a constant and defined product. Moreover, personalized treatments will also need large collections of phages to elaborate specific cocktails for each patient and for each type of infection. A tentative solution for this issue can be found in initiatives such as Phage Directory alerts (https://phage.directory/alerts (accessed on 15 June 2021)), which are sent worldwide when phages are needed to treat urgent infections. Nonetheless, in order to make phage therapy a reality, and to give a quick answer to the need for a particular phage, it is important that pure stocks of well-characterized phages, including sequence and host range data, are always available. Otherwise, there is not enough time to prepare the treatment for those patients that really need it. Although altruism works on some occasions, it cannot be expected for this solution to last in the long term. Therefore, this activity should be regulated and supported by the health authorities. In this context, PhageBank^TM^ is a growing collection of phages initiated in 2010 by the Biological Defense Research Directorate (BDRD) of the U.S. Naval Medical Research Center (NMRC) and Adaptive Phage Therapeuticals (APT company) in collaboration with the Mayo Clinic. Additionally, the availability of defined cocktails is also an approach that has been used for a long time in Eastern European countries and remains the most feasible option for many companies currently working on phage therapy.

Altogether, the results obtained so far regarding the ability of phages and phage proteins to treat pneumonia are promising, but still scarce. For the two main Gram-positive pathogens causing this disease, *S. pneumoniae* and *S. aureus*, most studies have been conducted in animals. As a result, proper clinical trials are still necessary prior to the implementation of these strategies in hospitals. This situation is one additional example of how new therapies need not only financial support to carry out the trials but also regulatory support to allow the use of new products, and the investment of manufacturing companies to put these products on the market. Overall, despite the multiple setbacks encountered when trying to treat and prevent pneumonia, we trust that the results obtained in the investigations that are currently underway will allow the development of effective therapeutic products against this disease.

## Figures and Tables

**Figure 1 antibiotics-10-01000-f001:**
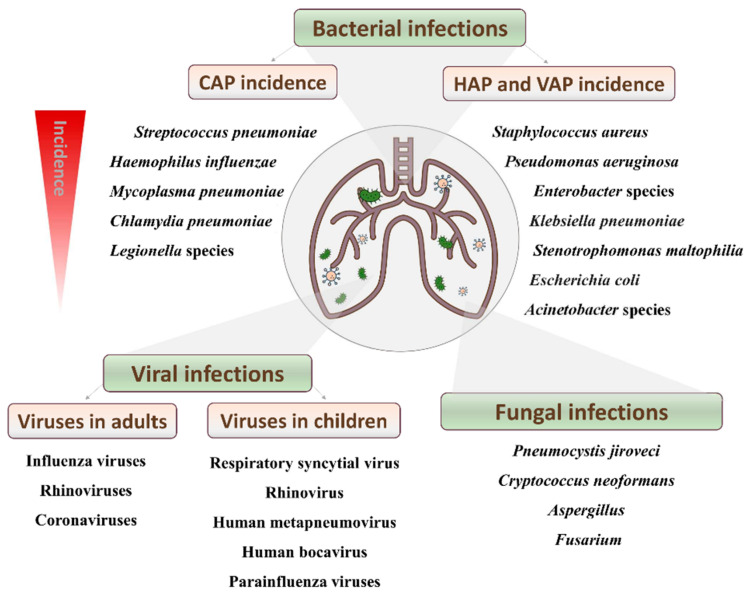
Main pathogens causing pneumonia and their incidence.

**Figure 2 antibiotics-10-01000-f002:**
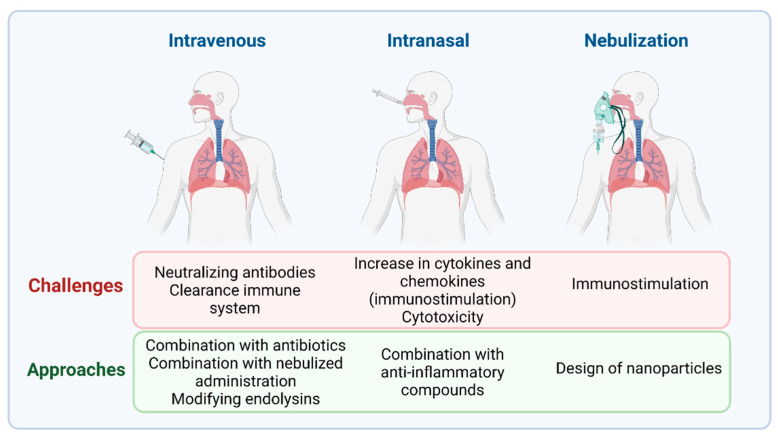
Different possibilities for delivery of phage therapy for the treatment of pneumonia caused by Gram-positive bacteria.

**Table 1 antibiotics-10-01000-t001:** State-of-the-art in treatments against pneumonia caused by Gram-positive bacteria.

Strategies	Effect	Examples	
Antibiotics	Bacterial killing or growth inhibition	Fluoroquinolones	Levofloxacin
			Moxifloxacin
		β-lactams	Amoxicillin
			Ampicillin
			Amoxicillin-clavulanate
			Ampicillin-sulbactam
			Cefotaxime
			Ceftriaxone
			Oxacillin
			Fluocloxacillin
			Ceftaroline fosamil
		Macrolides	Azithromycin
			Clarithromyxin
		Oxazolidinones	Linezolid
		Glucopeptides	Vancomycin
		Pleuromutilins	Lefamulin
Antibodies	Toxin neutralization		Ab pneumolysin
			Ab choline-binding PspA
		Monoclonal Ab	Human mAb IgG1 AR-301
			mAb MED14893
			2A3 and its variant LC10
			mAb LTM14
		Ab leukocidins	mAb ASN-1
			mAb ASN-2
Natural compounds	Toxin prevention	Flavonoids	Baicalin
			Morin hydrate
		Liposomes	
		Hydroxamate inhibitor	GI254023X
Antimicrobial peptides	Bacterial killing	AMPs	CATH-1
			CATH-2
			CRAMP
			LL-37
Immune modulators	Reduction inflammatory response	Toll-like	
		NOD-like	
Phages and phage proteins	Bacterial killing		Phages AB-SA01
			Pyophage
			Phage Sb-1
			Lysin Cpl-1
			Lysin Pal
			Lysin Cpl-7
			Lysin Cpl-711
			Lysin PL3
			Lysin ClyJ
			Lysin 23TH_48
			Lysin SA01_53
			Lysin SAL200

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
