# Peer review of "Gram-Positive Pneumonia: Possibilities Offered by Phage Therapy"

_antibiotics, 2021, doi:10.3390/antibiotics10081000_

Round 1

Reviewer 1 Report

The manuscript by Fernadez et al offers a well-rounded review of bacterial pneumonia caused by Streptococcus pneumoniae and Staphylococcus aureus. The review does a good job of introducing pneumonia and the numerous causative agents (covering bacterial, viral and fungal) before specifically focusing on bacterial pneumonia primarily caused by two gram-positive pathogens above. The review then goes on to detail current therapeutic options available (both preventative and therapeutic) before introducing the concept of phage therapy and details its advantages of current options and the breadth of research results in this area. The manuscript is well written, and the figures are detailed and understandable. I have only the following minor comments:

Line: 246- (Table 1) The formatting of Table 1, especially as it is split over 2 pages, can be a bit jarring to interpret upon first viewing. This is mostly due to the rapid change from double column tabulation in the Examples column when transitioning from Antibodies to Antibiotics on the second page of the Table. It is possible to standardise Ab pneumolysin and Ab choline-binding protein PspA into a double column format?  The single column formatting works fine for the Phages and phage proteins section, as it is more substantial. In regards to the phage section, it appears the label Phages and protein proteins is incorrectly “centered” (i.e. it is not in line with “Bacterial killing”) and not consistent with the formatting of the rest of the table.   

Line 332-333. Given the reference here is from 2004, I wonder if the authors could expand a little on the state of S. pneumoniae bacteriophage landscape in the last two decades. If there is little work done in this area in the isolation of S. pneumoniae phages, perhaps the authors could explain to the reader why.  

The study by Wang et al (2020) Can bacteriophage endolysins be nebulised for inhalation delivery against Streptococcus pneumoniae?, may be appropriate for inclusion in this review.

Author Response

Reviewer 1

The manuscript by Fernadez et al offers a well-rounded review of bacterial pneumonia caused by Streptococcus pneumoniae and Staphylococcus aureus. The review does a good job of introducing pneumonia and the numerous causative agents (covering bacterial, viral and fungal) before specifically focusing on bacterial pneumonia primarily caused by two gram-positive pathogens above. The review then goes on to detail current therapeutic options available (both preventative and therapeutic) before introducing the concept of phage therapy and details its advantages of current options and the breadth of research results in this area. The manuscript is well written, and the figures are detailed and understandable. I have only the following minor comments:

Line: 246- (Table 1) The formatting of Table 1, especially as it is split over 2 pages, can be a bit jarring to interpret upon first viewing. This is mostly due to the rapid change from double column tabulation in the Examples column when transitioning from Antibodies to Antibiotics on the second page of the Table. It is possible to standardise Ab pneumolysin and Ab choline-binding protein PspA into a double column format?  The single column formatting works fine for the Phages and phage proteins section, as it is more substantial. In regards to the phage section, it appears the label Phages and protein proteins is incorrectly “centered” (i.e. it is not in line with “Bacterial killing”) and not consistent with the formatting of the rest of the table.   

Thank you very much for your suggestions. The table has been modified accordingly.

Line 332-333. Given the reference here is from 2004, I wonder if the authors could expand a little on the state of S. pneumoniae bacteriophage landscape in the last two decades. If there is little work done in this area in the isolation of S. pneumoniae phages, perhaps the authors could explain to the reader why.  

The text has been modified accordingly.

The study by Wang et al (2020) Can bacteriophage endolysins be nebulised for inhalation delivery against Streptococcus pneumoniae?, may be appropriate for inclusion in this review.

Thank you very much for your suggestions. The text has been modified accordingly.

Reviewer 1

The manuscript by Fernadez et al offers a well-rounded review of bacterial pneumonia caused by Streptococcus pneumoniae and Staphylococcus aureus. The review does a good job of introducing pneumonia and the numerous causative agents (covering bacterial, viral and fungal) before specifically focusing on bacterial pneumonia primarily caused by two gram-positive pathogens above. The review then goes on to detail current therapeutic options available (both preventative and therapeutic) before introducing the concept of phage therapy and details its advantages of current options and the breadth of research results in this area. The manuscript is well written, and the figures are detailed and understandable. I have only the following minor comments:

Line: 246- (Table 1) The formatting of Table 1, especially as it is split over 2 pages, can be a bit jarring to interpret upon first viewing. This is mostly due to the rapid change from double column tabulation in the Examples column when transitioning from Antibodies to Antibiotics on the second page of the Table. It is possible to standardise Ab pneumolysin and Ab choline-binding protein PspA into a double column format?  The single column formatting works fine for the Phages and phage proteins section, as it is more substantial. In regards to the phage section, it appears the label Phages and protein proteins is incorrectly “centered” (i.e. it is not in line with “Bacterial killing”) and not consistent with the formatting of the rest of the table.   

Thank you very much for your suggestions. The table has been modified accordingly.

Line 332-333. Given the reference here is from 2004, I wonder if the authors could expand a little on the state of S. pneumoniae bacteriophage landscape in the last two decades. If there is little work done in this area in the isolation of S. pneumoniae phages, perhaps the authors could explain to the reader why.  

The text has been modified accordingly.

The study by Wang et al (2020) Can bacteriophage endolysins be nebulised for inhalation delivery against Streptococcus pneumoniae?, may be appropriate for inclusion in this review.

Thank you very much for your suggestions. The text has been modified accordingly.

Reviewer 2 Report

The review manuscript is well written and does a nice job summarising the incidence of gram positive bacteria in pneumonia cases and the challenges of diagnosing and treating the disease. Minor suggestions are made regarding a couple of points made in the conclusions and perspectives section. Additionally, while Figure 2 is an attractive image, in the current form doesn't seem to add to the intended point (ie challenges selecting delivery method). Additional information about pros/cons/challenges/concerns or similar info would add value to the manuscript.  

See specific comments on the attached pdf.

Author Response

Reviewer 2

The review manuscript is well written and does a nice job summarising the incidence of gram positive bacteria in pneumonia cases and the challenges of diagnosing and treating the disease. Minor suggestions are made regarding a couple of points made in the conclusions and perspectives section. Additionally, while Figure 2 is an attractive image, in the current form doesn't seem to add to the intended point (ie challenges selecting delivery method). Additional information about pros/cons/challenges/concerns or similar info would add value to the manuscript.  

For example, aerosolized phages are more effective than those used by intravenous administration, as the latter would be more likely to stimulate the production of neutralizing antibodies. Reference

Done

Figure 2. Text refers to the challenges of selecting the delivery method. Additional information regarding these challenges would be valuable

Figure 2 has been modified.

Currently, most of the phage products used in the hospitals of Western countries are produced as magistral preparations, which are formulated on demand as personalized medicines. This is a drawback for companies that need to have a constant and defined market.

Did you mean a defined product? Suggest you expand on the defined cocktail approach to balance the idea of personalized therapy. Are the approaches complementary?

Thank you very much for your suggestions. The text has been modified accordingly.

This situation is one additional example of how new therapies need not only financial support to carry out the trials, but also regulatory support to encourage interested companies to invest in these products.

Suggest revision of this statement, cause and effect description is a rather contrived by suggesting regulatory agencies have direct power to drive investment. Futhermore, clinical data is required to change regulations not just money

Thank you very much for your suggestions. The text has been modified accordingly.